# Estimation of Viscosity and Yield Stress of Cement Grouts at True Ground Temperatures Based on the Flow Spread Test

**DOI:** 10.3390/ma13132939

**Published:** 2020-06-30

**Authors:** Zhipeng Xu, Yichen Miao, Haikuan Wu, Xun Yuan, Changwu Liu

**Affiliations:** 1College of Water Resource and Hydropower, Sichuan University, Chengdu 610065, China; wuhaikuan@stu.scu.edu.cn; 2State Key Laboratory of Hydraulics and Mountain River Engineering, Sichuan University, Chengdu 610065, China; 3Institute for Disaster Management and Reconstruction, Sichuan University-The Hong Kong Polytechnic University, Chengdu 610207, China; miaoyichen@stu.scu.edu.cn (Y.M.); xunyuantyut@gmail.com (X.Y.)

**Keywords:** viscosity, yield stress, cement grout, flow spread test, temperature, deep rock grouting

## Abstract

The rheology of cement grouts often plays a crucial role in the success of rock grouting. In practice, the rheological parameters should be timely adjusted according to the evolution of grouting pressure, flow rate and injection time. However, obtaining the magnitude of rheological parameters is not easy to achieve under site conditions. More importantly, the ground temperature in deep rock masses is elevated higher than that on the surface or under room conditions, which has been demonstrated to strongly influence the rheological properties of grouts. Reasonable understanding and control of the rheological behavior of cement grouts at true ground temperatures is very important to the quality of grouting. This paper aims to propose a simplified method to approximately estimate the initial yield stress and viscosity of cement grouts for rock grouting under elevated ground temperature that actually exists in deep rock masses, on the basis of the flow spread test. The temperature investigated was controlled between 12 °C and 45 °C to simulate the true ground temperature in rock masses with a maximum depth of 1500 m below the surface. Taking the influences of elevated temperatures into account, a temperature-based model for estimating the initial viscosity of cement grout was successfully developed on the basis of Liu’s model and the results of the flow spread test. However, the yield stress failed to be estimated by the Lapasin model due to the absence of plastic behavior of cement grouts. In contrast, yield stress can be linearly correlated to the measured relative flow area. In this work, it was also found that the dependence of yield stress of cement grouts on relative flow area is a strongly exponential law. The temperature dependence of the viscosity of water was accounted for in both estimations of viscosity and yield stress of grouts. Significantly, it was found that the packing density of cement is dependent on the grout temperature, especially when the temperature is up to 45 °C. The proposed method in this work offers an alternative solution for technicians to reasonably control the rheological properties in the increasing applications of deep rock grouting.

## 1. Introduction

The rheology of cement grouts can significantly influence the workability during grouting and consequently the sealing quality that will be achieved in rock masses to be grouted [1,2,3]. Because of the varying impacts of cement hydration and thixotropic behavior, the rheology of cement grouts behaves in a strong time dependent manner, and is quite complex and difficult to measure or define [1]. However, two parameters could be employed to characterize the rheological properties of cement based grouts, namely yield stress and plastic viscosity [4]. Yield stress is the resistance of vicious grout that must be overcome to initiate the flow, which is often utilized to estimate the maximum grout spread, i.e., the penetration distance [5,6]. Plastic viscosity represents the effect of internal frictions which will strongly affect the flow of grout in a medium. These two rheological parameters are of great significance to the design of qualified grout mixtures due to the grout quality determining the success of grouting to a large extent. Unfortunately, there is still no standard method to determine the values of yield stress and viscosity [6]. The most common approach is to measure the evolution of shear stress versus shear rate using a rotational viscometer in the laboratory, and then to obtain the yield stress and viscosity using fitted fluid models including Bingham or modified Bingham, or Herschel–Bulkely models [5]. The process above requires lots of precision measurements in the laboratory and skilled investigators. For most grouting engineers, it is often very difficult to achieve this at the jobsite. In fact, the rheological properties of grouts should be timely adjusted according to the evolution of grouting pressure, flow rate and grouting time [7]. Although the outflow time of the March cone could provide some information associated with the flowability of grouts, it cannot directly offer any rheological parameters [8]. Considering the importance of rheological parameters for grouting control, there is a need to establish an applicable approach that can be performed on site by field technicians to readily estimate the yield stress and viscosity of cement grouts.

Conventionally, the rheological behavior of cement grouts is usually influenced by multiple factors including the water–cement ratio, the fineness of the cement, additives and the mixing process [9]. In general, the water–cement ratio has the greatest impact on the rheological properties of cement grouts [10]. Specifically, the fluidity of grout often increases with an increasing water content, while setting time and bleeding of cement grouts decrease with an increasing water–cement ratio. More importantly, the viscosity of cement grouts will be considerably enhanced with the decrease of dosage of water. The grain size of cement, i.e., the fineness characteristics of cement particles, also influences the flow properties of grouts at fresh state and mechanical behavior at hardened state [11]. When the rheology of grouts is not satisfying, additives such as bentonite, water glass, fly ash and other chemical compounds are utilized to improve the rheological performance of grouts [10]. Great attention has been devoted to the rheological properties of cement grouts at room conditions [1,3,10,12,13,14,15,16]. With increasing grouting applications performed at coal regions or in deep underground conditions in recent years, the ambient temperature is considered as another critical factor that affects the yield stress and viscosity of cement grouts, owing to the effects of temperature on the hydration rate of cement [9,17,18,19,20]. The temperature of underground formations usually varies proportionally with an increase in depth, which is often much higher than on the surface [21]. The ground temperature in underground rock masses will be increased from about 30 °C at a depth of 800 m to 40 °C at a depth of 1000 m, and when the depth is up to 1500 m, the maximum ground temperature will be elevated to approximately 50 °C [22]. With increasing deep mining, the corresponding depth of rock mass to be injected accordingly becomes deeper and deeper. Thus, the impact of ground temperature must be taken into consideration in the determination of the rheology of cement grouts. On the other hand, it has been found that the viscosity of concentrated suspensions closely depends on the relative concentration of the mixture. Investigators including Mooney [23], Krieger and Dougherty [24], Chong [25], Dabak and Yucel [26], and Liu [27] have developed some theoretical/empirical models to predict the relative viscosity of suspensions (i.e., the ratio of measured viscosity of the suspension to the viscosity of the continuous phase). Two parameters are always involved in these models, namely the packing density and volume fraction of solid powders. Packing density is the maximum solid proportion corresponding to a water-saturated fresh mixture [28]. There are generally three parameters correlated to temperature: the viscosity of the suspension, the viscosity of the continuous phase and the packing density. The effect of ground temperature is not included in these models—different depths will have different ground temperatures, which will influence the rheological behavior of cement grouts. Accordingly, the viscosities of the disperse and continuous phases will be subject to the impacts of elevated temperature in deep rock masses. Moreover, although several theoretical approaches have been developed to determine packing density, the effect of temperature seems to be left out [29,30]. Since the flow properties of grouts will be considerably influenced by temperature, the packing density of cement at various temperatures might be different. Elevated temperature may also cause potential impacts to determining the packing density.

The overall objective of this work is to develop a simplified method to predict the viscosity and yield stress of cement grouts, considering the effect of elevated ground temperature on the rheology of both cement grouts and water. Temperature is controlled between 12 °C and 45 °C to simulate the true ground temperature in rock masses with a maximum depth of 1500 m below the surface. The viscosities of cement grouts with different water–cement ratios are experimentally measured using a rotational viscometer. Based on the results of the flow spread test developed by Okamura [31], the existing model, i.e., Liu’s model, is combined with the effect of temperature on the continuous phase (i.e., water), to estimate the viscosity of cement grout. Both the Lapasin model and linear correlation are used to model the yield stress of cement grouts. The packing density of cement at various temperatures is also determined by the flow spread test. It was found that the viscosity and yield stress of cement grouts at different temperatures can be readily predicted using the proposed method.

## 2. Materials and Methods

### 2.1. Experimental Investigation

#### 2.1.1. Materials

The cement grouts used are the same as those used in the rock grouting of Chinese coal mines, with water–cement ratios (w/c) of 0.5–1.5. Usually, the cement grout consists of cement, water and chemical additives including sodium chloride (NaCl) and triethanolamine (TEA). Among them, the cement used is Xi-Nan 42.5 ordinary Portland cement produced in Sichuan by China National Building Material Group Co., Ltd., according to the Chinese standard GB 175-2007 [32]. Table 1 shows the specific characteristics of the cement provided by the manufacturer, and the grain size distribution measured using a laser diffraction particle size analyzer is represented in Figure 1. The water used for the grout mixture preparation is tap water.

Sodium chloride (NaCl) as an accelerator and triethanolamine (TEA) as a gas-producing agent are used as chemical additives to improve the performance of cement grouts at fresh state. NaCl used in this work is in the form of small particles, with a purity of 99.5%. TEA utilized in this work is produced by the Dow Chemical company (Midland, MI, USA), with a purity of 99%. Because of the relatively low dosage of TEA, it is firstly diluted with fresh water to a concentration of 5 wt.% in order to accurately control the content of TEA to be added into the cement grout mixture. The water used as the solvent in the diluted TEA solution must be counted into the total water.

#### 2.1.2. Proportioning and Mixing

In order to study the effect of temperature on the rheology of cement grouts, all materials involved in the grout mixtures, including cement, water and additives, should be controlled at the desired temperatures. For better control of grout temperature, the cement to be used is warmed in a climatic chamber for at least 8 h to obtain the desired cement temperature, whereas water and diluted TEA solution are heated using an IKA magnetic stirrer, as shown in Figure 2. Grout samples are prepared at temperatures of 12 °C, 25 °C, 35 °C and 45 °C to simulate the true ground temperatures at different depths. Proportioning of cement grout mixtures is shown in Table 2.

During mixing, cement is first mixed with the solid NaCl while the TEA solution is mixed with water. Then, the dry powders are added into aqueous solution and mixed for 5 min at 500 rpm using a digital agitator, as shown in Figure 3. The grout temperature is precisely control by the IKA magnetic stirrer. For the two grouts with a w/c of 0.5 and 0.6, the mixing speed is 800 rpm because of their thick states.

#### 2.1.3. Flow Spread Test

The initial viscosity of grouts strongly depends on the volumetric percentage of solid particles (i.e., cement). To correlate the solid content to the viscosity of grout, the packing density of cement must be determined in advance. Meanwhile, the effect of temperature on packing density should be clarified by laboratory testing. Therefore, in this work, the packing density of solid particles ($\emptyset_{m}$) under various temperatures is determined by the flow spread test, which is developed by Okamura [31]. The apparatus for the flow spread test is relatively simple, consisting of a metal frustum cone and a tempered glass plate, as shown in Figure 4. The height of the frustum cone is 60 mm, the diameter at the top is 70 mm, and the diameter at the base is 100 mm. The glass plate is placed horizontally and then checked using a leveling instrument, as shown in Figure 5a. In order to reduce the impact of the temperature of the cone, the frustum cone is immerged in hot water at the same temperature as that of the grout to be tested, and then dried with clean tissues before testing. During testing, the cone is placed at the center of the glass plate and filled with the grout mixture. Immediately after filling, the cone is lifted upright in 1 s, and thus the grout will spread freely over the glass plate, as shown in Figure 5b. After 30 s, the average diameter (*D*) of the spread can be measured and thus the relative flow area (*R*) of the grout can be determined by Equation (1).
(1)$$R=\frac{\left(D^{2}-100^{2}\right)}{100^{2}}={\left(\frac{D}{100}\right)}^{2}-1$$

It is reported that the relative flow area *R* calculated is linearly related to the volume ratio of water to solid powder ($\frac{V_{water}}{V_{powder}}$) in suspension, and then the water demand of cement ($\beta_{p}$) at specific temperatures could be estimated by Equation (2) [31]. Then, the packing density of cement $\emptyset_{m}$ at specific temperatures could be determined using Equation (3) [28].
(2)$$\frac{V_{water}}{V_{powder}}=\beta_{p}+RE_{p}$$
(3)$$\emptyset_{m}=\frac{1}{1+\beta_{p}}$$
in which $E_{p}$ is the slope of fitted line.

As the spread of grouts is influenced by the initial flowability of grouts, which might vary with grout temperature, flow spread tests are conducted at temperatures of 12 °C, 25 °C, 35 °C and 45 °C, respectively. The deviation of grout temperature is no more than ±1 °C. Cement grouts are tested within 2 min after a 5-min mixing to avoid possible influences caused by grout segregation (i.e., the bleeding).

#### 2.1.4. Viscosity Measurement

The initial viscosity of cement grout is measured by using a NXS-11B concentric cylinder rotary viscometer (made by Chengdu instrument factory in Chengdu, China). During testing, the grout temperature is precisely controlled at the predefined value by the thermostatic bath, as shown in Figure 6. The torque representing the shear stress imposed to the grout is measured at a given shear rate (i.e., rotation rate). Based on the measured values, the apparent viscosity can be determined by Equation (4).
(4)$$\eta =\frac{\tau }{\dot{\gamma }}$$
in which *η* denotes the dynamic viscosity, *τ* is the measured shear stress and $\dot{\gamma }$ is the shear rate.

The shear rate is gradually increased from 0 to 200 s^−1^ or to 996 s^−1^, depending on the concentration of grout. For each step, the shear rate is kept constant for 10–15 s to obtain an equilibrium state. Viscosity measurements should be performed within 2 min after mixing. As for the flow spread test, the viscosity measurements are conducted at temperatures of 12 °C, 25 °C, 35 °C and 45 °C, respectively.

### 2.2. Modelling of Water Viscosity

As a typical Newtonian fluid, the viscosity of liquid water is highly dependent on temperature [33]. In general, the viscosity of water decreases with the increase in temperature [34]. The dependence of the viscosity of a liquid with relatively large intermolecular forces on temperature can be described by Arrhenius’ law [35], as shown in Equation (5).
(5)$$\eta =Aexp\left(\frac{\Delta E_{vis}}{RT}\right)$$
where *A* is a constant, $\Delta E_{vis}$ is the fluid-flow activation energy, and *T* is the fluid temperature in unit *K*. *R* is the ideal gas constant, equal to 8.314 J·mol^−1^.K^−1^.

As the viscosity of liquid water is very low and difficult to measure without a high resolution viscometer, potential data available in the literature is utilized to determine the dependence of the viscosity of water on temperature on the basis of Equation (5).

### 2.3. Estimation of Viscosity and Yield Stress

#### 2.3.1. Estimation of Viscosity

For concentrated suspensions, the relative viscosity always highly correlates to the solid fraction ∅ at the 95% confidence level, regardless of shear rate [36]. Existing models describing the correlations include the Krieger–Dougherty (K–D) equation, Chong et al.’s model, Liu’s model, Mooney’s equation and Dabak et al.’s model, as shown in Table 3 [37]. 

Taking the varying viscosity of water at specific temperatures, the relative viscosity $\eta_{r}$ can be obtained by Equation (6).
(6)$$\eta_{r}=\frac{\eta_{suspension}}{\eta_{continuous}}=\frac{\eta_{suspension}}{Aexp\left(\frac{\Delta E_{vis}}{RT}\right)}$$
where $\eta_{suspension}$ is the measured apparent viscosity at a specific shear rate under temperature *T*. 

In the study conducted by Senapati et al. [37], it is found that Liu’s model can better describe the evolution of viscosity versus solids content when the volume fraction of solids in suspension is below 40%. In this work, the maximum concentration of solids in the thickest grout (w/c = 0.5) is 0.396, which is not more than 0.4. Therefore, the estimation of viscosity of cement grouts at different temperatures is based on Liu’s model.

#### 2.3.2. Prediction of Yield Stress

The yield stress of suspensions, including cement grouts, generally increases with increasing solid volume fraction [36]. For suspensions, the correlation between solid content and the yield stress can be described by the Lapasin et al. model, as shown in Equation (7) [38,39].
(7)$$\tau_{0}=K{\left(\frac{\emptyset -\emptyset_{0}}{\emptyset_{m}-\emptyset }\right)}^{m}$$
in which $\tau_{0}$ is the yield stress, *K* and *m* are constants and $\emptyset_{0}$ is the lower limit of the solid volume fraction at which the shear thinning is transitioned to plastic flow behavior.

On the other hand, Domone et al. [31] found that the rheological parameters of fresh suspensions, including yield stress, are linearly correlated to the results of the flow spread test. In this case, it is possible to predict the yield stress of fresh cement grout if the linear relationship can be confirmed by experimental results.

Therefore, both methods are utilized to estimate the initial yield stress of cement grouts at different temperatures. The reliability of the two approaches will be analyzed if both are available.

## 3. Results and Discussion

### 3.1. Evolution of Packing Density $\emptyset_{m}$

According to the results of the flow spread test, the evolution of the relative flow area *R* of cement grouts at different temperatures is shown in Figure 7. In general, the relative flow area decreases slightly with increasing temperature. To evaluate the discrepancy between measurements, the deviation of measured values from the average relative flow area is defined as Equation (8).

(8)$$D=\frac{\left|R_{T}-R_{avg}\right|}{R_{avg}}\times 100\%$$ in which $R_{T}$ is the measured relative flow area of a specific grout mixture at temperature *T* and $R_{avg}$ is the average value at all temperatures investigated. From Figure 8, it could be found that except for the thick cement grouts (w/c ≤ 0.6), most measurements of the flow spread test have a small deviation of less than 10%. Considering the error caused by the experimental operations, the effect of elevated temperature on the relative flow area of thinner grouts (w/c ≥ 0.8) is quite slight. However, for thick cement grouts with water–cement ratios of 0.5 and 0.6, the relative flow area decreases with increasing temperature, especially when the temperature is up to or beyond 35 °C. This is because the higher temperature can deteriorate the initial rheology of cement grouts, i.e., increased viscosity at high temperature will enhance the resistance to flow, which is in agreement with the literature [18].

Following Okamura et al. [31], the water demand of cement $\beta_{p}$ at a given temperature can be determined by the linear fitting between *R* and solid content. As shown in Figure 9, the linear relationship between the relative flow area and solid fraction in volume is very evident. Calculated values of water demand and packing density at different temperatures are shown in Table 4 and Table 5, respectively. It can be seen that the water demand of cement $\beta_{p}$ increases with increasing temperature, while the packing density of cement $\emptyset_{m}$ is inversely proportional to increasing temperature. The maximum packing density is 0.641 which is observed at 12 °C, whereas the minimum value is 0.505 obtained at 45 °C.

The temperature dependence of packing density is shown in Figure 10. There seems to be a linear correlation between the packing density and temperature, and thus the linearly fitted model is given in Equation (9). Although the correlation coefficient obtained is not as satisfying as expected due to very limited data, the fitted model still can give valuable information about the evolution of packing density at temperatures ranged from 10–50 °C. On an engineering level, the effect of temperature on the packing density of cement across the investigated range of temperatures can be described by Equation (9) with appropriate accuracy. In this work, we suppose the relationship is linear, as shown in following equation.
(9)$$\emptyset_{m}=0.72-0.0045T$$

### 3.2. Measured Viscosity and Yield Stress

The initial measured apparent viscosities of cement grouts with ratios of 0.5, 0.6, 0.75, 0.8, 1.0, 1.2 and 1.5 at different temperatures are given in Figure 11, Figure 12, Figure 13 and Figure 14. The apparent viscosity of thick cement grouts (w/c ≤ 0.8) is heavily influenced by elevated temperature, especially when the temperature is higher than 35 °C. For thinner grouts (w/c ≥ 1.0), the variation of apparent viscosity versus temperature is very slight. Shear thinning behavior could be observed in thick cement grouts (w/c ≤ 0.8). Thick grouts (w/c ≤ 0.8) at different temperatures behave as typical non-Newtonian fluids, while the rheological behavior of thinner grouts (w/c ≥ 1.0) can be appropriately described by the Newtonian fluid model because of the very few changes in viscosity caused by shear rates. This is in agreement with the literature [40]. On the other hand, the effect of temperature on the viscosity variation is much more pronounced in thick cement grouts (w/c ≤ 0.8) than that in thinner grouts (w/c ≥ 1.0). Thus, the viscosity evolution of thick cement grouts should be focused on because of the large changes in viscosity at different shear rates. As a result, the estimation of viscosity of thick cement grouts (w/c ≤ 0.8) becomes the aim of this work.

Additionally, the elevated temperature does not change the trend of viscosity variation, which means that at the investigated temperatures, the constitutive relationship of cement grouts with a specific water–cement ratio can be properly described by a fixed fluid model, such as the Bingham model or the Herschel–Bulkley model. It offers strong support to the estimation of the final penetration of grout under specific conditions in which the fluid model can be treated as a fixed fluid model. However, the time dependence of the rheology of grouts should not be neglected, as it may considerably influence the flow performance of grouts, but this is out of the scope of this work.

It is reported that in practice, the actual shear rate of grouts during injection is often between 200 s^−1^ and 300 s^−1^ [41]. Accordingly, the evolution of viscosity at shear rates in this range should be focused on by technicians. Therefore, the apparent viscosity of thick grouts (w/c ≤ 0.8) at about 200 s^−1^ is selected to establish the estimation of viscosity, as shown in Table 6. The modified Bingham fluid model [42] is employed to determine the experimental values of yield stress, as illustrated in Table 7.

### 3.3. Temperature Dependence of Water Viscosity

The viscosity of water is temperature dependent [43]. For example, the viscosity of water at 15 °C is about 1.1383 mPa·s, while at 40 °C it is reduced to 0.6527 mPa·s [33]. Although the variation of absolute values of water at different temperatures is quite small, the deviation of the viscosity of water at 40 °C is up to 43% of that at 12 °C. In estimating the viscosity of grouts, the relative viscosity $\eta_{r}$ is defined as the ratio of grout viscosity to the viscosity of the corresponding liquid phase (water), i.e., $\eta_{r}=\frac{\eta_{grout}}{\eta_{water}}$. Consequently, the relative viscosity of grout mixtures will be considerably underestimated at higher temperatures without consideration for the influence of reduced water viscosity. The great deviation of viscosity of water at different temperatures must affect the relationship between grout viscosity and solid content. Therefore, the temperature dependence of water viscosity must be defined properly.

The ground temperature in underground rock masses with a maximum depth of about 1500 m is usually up to 45 °C or higher [22]. Hence, a correlation between the viscosity of water and temperature should be in the range of investigated temperatures. Since the task of precisely measuring the viscosity of water is quite difficult, the experimental results available in the literature (reported by Ma and Yuan [44]) is utilized to determine the temperature dependence of the viscosity of water, as shown in Table 8. When the temperature is changed within a small range, the fluid-flow activation energy $\Delta E_{vis}$ can be approximated to a constant according to ASTM E698-05 [45]. In the range of 0–100 °C, Arrhenius’ equation is fitted with the experimental viscosity of water, as shown in Figure 15. The correlation of water viscosity and temperature is given in Equation (10), in which $T_{k}$ is temperature unit *K*.
(10)$$\eta_{w}=0.00108e^{\frac{2016}{T_{k}}}$$

### 3.4. Prediction of Viscosity at True Ground Temperature

The estimation of the viscosity of grouts at true ground temperature is based on Liu’s model. Liu [27] proposed a general form of the viscosity prediction model for both lowly and highly concentrated suspensions, as shown in Equation (11). To predict grout viscosity $\eta_{g}$, the equation can be also expressed in the form of Equation (12).
(11)$$\eta_{r}={\left[a\left(\emptyset_{m}-\emptyset \right)\right]}^{-n}$$
(12)$$\eta_{g}=\eta_{w}{\left[a\left(\emptyset_{m}-\emptyset \right)\right]}^{-n}$$
in which *a* is a constant to be determined and *n* is a flow-dependent parameter, both of which can be determined by the grout viscosity $\eta_{g}$/solid fraction ∅ relation. Substituting Equations (6), (9) and (10) into Equation (12), the apparent viscosity of grout can be expressed as:(13)$$\eta_{g}=0.00108e^{\frac{2016}{273+T}}{\left[a\left(0.72-0.0045T-\emptyset \right)\right]}^{-n}$$

The measured viscosity of cement grouts at w/c = 0.75 was selected to be predicted; other results are utilized to determine the constants of *a* and *n*. The fitted correlation of viscosity and the volume fraction of cement is shown in Figure 16, and the obtained values of *a* and *n* are listed in Table 9. Comparing the values of *n* obtained in this work with that reported in the literature [27,46], it can be found that the magnitude of *n* in cement grouts is much greater than that in highly concentrated cement suspensions. The predicted viscosity of cement grouts at w/c = 0.75, i.e., ∅=0.304183, is shown in Table 10. In general, the estimation is over the measured value by 15–30%. The degree of deviation is close to that found in the literature conducted with the K–D model [46]. In this case, the initial viscosity of cement grouts can be appropriately estimated using the proposed method, considering the impacts of ground temperature.

### 3.5. Estimation of Yield Stress

In the prediction model for yield stress proposed by Lapasin et al. [38], the variables of *K*, $\emptyset_{0}$, $\emptyset_{m}$ and *m* are flow-dependent. Therefore, they are also temperature dependent. At a given temperature *T*, the values of *K*, $\emptyset_{0}$ and *m* can be determined by data regression analysis. For this work, all Lapasin-based model fittings failed, indicating that the yield stress of cement grouts cannot be correlated to the solids content via the Lapasin model. The possible cause for this is that the cement grouts with the most common water–cement ratios behave as a typically shear thinning fluid, but no plastic behavior can be observed. In the Lapasin model, there is a transition in fluid performance from shear thinning to plastic behavior [39], resulting in an S-shaped curve of the volume fraction of solids versus the initial yield stress, as shown in Figure 17. However, the approximately linear relationship between the yield stress of cement grouts and the solids content cannot be evaluated by the Lapasin model due to the absence of plastic behavior of cement grouts.

In this case, the linear correlation is utilized to evaluate the relative flow area dependence of the yield stress of cement grouts [31]. Since the flow properties of cement grouts largely depend on the temperature, the correlation of yield stress versus relative flow area should be influenced by grout temperature. Figure 18 shows the relationship between the yield stress and relative flow area at various temperatures, which is approximately linear. Fitted models to describe the correlation of yield stress and relative flow area are given in Table 11. The yield stress generally decreased with increasing relative flow area, and high temperature has a much more pronounced influence on yield stress. This result agreed with the expectation, i.e., the yield stress of cement grouts at higher temperature is much higher than that at lower temperatures, which inevitably causes a smaller spread range. In this work, the correlation coefficient *R*^2^ at lower temperatures is not sufficiently satisfying, owing to the experimental errors and number of measurements. However, the linear relationship between the yield stress and relative flow spread can be expected and thus the yield stress could be estimated by simple testing, i.e., using the flow spread test.

Additionally, it can be found that the variation in yield stress seems to be an exponential function of relative flow area. Therefore, we attempted to fit the yield stress with relative flow area using an exponential function in the form of $f\left(x\right)=ab^{x}$, in which *a* and *b* are two constants to be determined. As shown in Figure 19, the evolution of yield stress very highly correlates with relative flow area in exponential law. The fitted exponential models are shown in Table 12, with considerably high correlation coefficients. Compared with linear correlation, it can be found that the exponential model has greater reliability to predict the yield stress of cement grouts. However, it should be noted that whether the exponential model can apply to other cement grouts with a greater range of mixing ratios needs to be examined further because of the limited experiments in this study. We have shown that it is applicable for estimating the yield stress of cement grouts with ratios ranging from 0.5 to 0.8.

## 4. Conclusions

The initial viscosity and yield stress of cement grouts with common water–cement ratios were targeted to be estimated based on the results of the flow spread test, considering the effect of true ground temperature in deep rock masses. The conclusions are summarized below:(1)The packing density of cement, determined by flow spread test, was temperature dependent. It generally decreases with increasing temperature. Hence, the effect of temperature on packing density should be taken into account in related issues. The temperature dependence of packing density was found to be linear in this work.(2)The initial viscosity and yield stress of thick grouts (w/c ≤ 0.8) were prone to be improved by elevated temperature. The rheology of thick cement grouts should be focused on more in deep rock grouting.(3)If the fluctuation of water viscosity at different temperatures was not taken into account, the relative viscosity of cement grout mixtures was considerably underestimated at higher temperatures, resulting in unreasonable understanding of the viscosity of cement grouts at true ground temperatures.(4)Based on Liu’s model and the flow spread test, a temperature-based model for estimating the initial viscosity of cement grout was successfully developed. In the proposed prediction model, the effects of elevated temperature on both water viscosity and the packing density of cement were properly taken into account. The developed method for predicting the viscosity of cement grouts produced sufficient accuracy at the engineering level, which will facilitate field technicians to readily control the viscosity of cement grouts at true ground temperatures in deep rock grouting.(5)The yield stress of cement grouts cannot be predicted using the Lapasin model due to the absence of plastic behavior of cement grouts. In contrast, it was linearly correlated to the results of the flow spread test, i.e., the relative flow area. In addition, it was also found that the dependence of the yield stress of cement grouts on the relative spread area is in the strongly exponential law in form of $f\left(x\right)=ab^{x}$ with the highest reliability for the estimation of yield stress of the investigated cement grouts.


## Figures and Tables

**Figure 1 materials-13-02939-f001:**
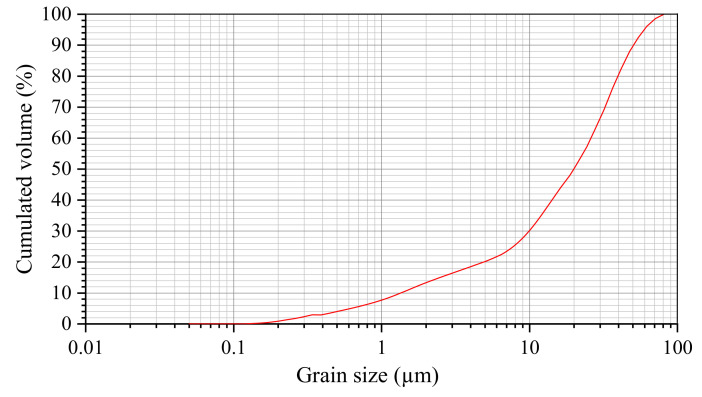
Grain size distribution of the cement.

**Figure 2 materials-13-02939-f002:**
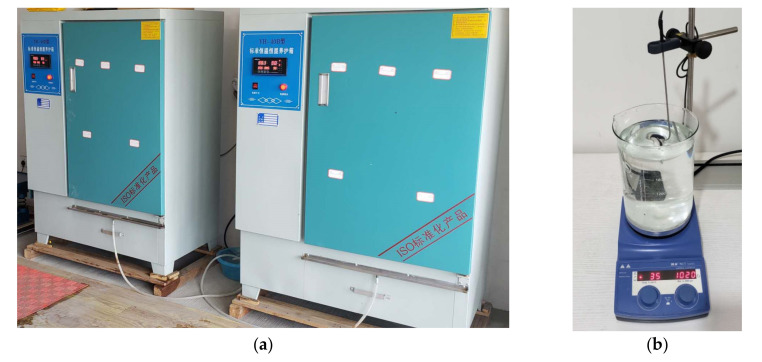
Temperature control of raw materials. (**a**) Cement heated using the climatic chamber. (**b**) Water heated using the IKA magnetic stirrer.

**Figure 3 materials-13-02939-f003:**
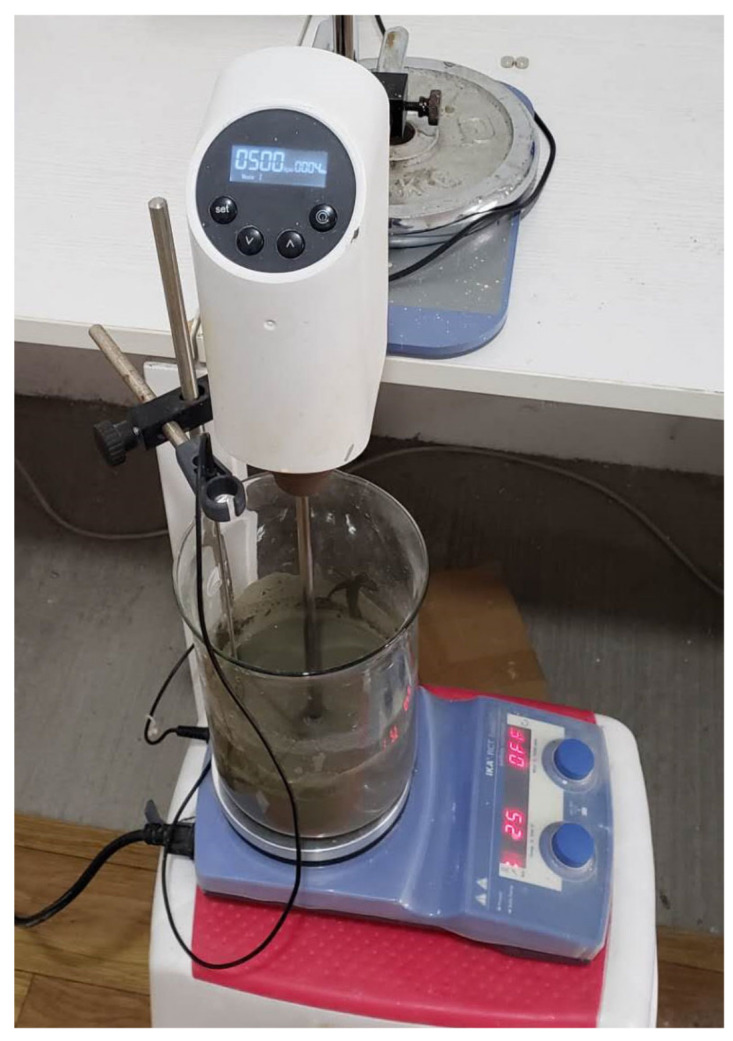
Mixing and temperature control of grouts.

**Figure 4 materials-13-02939-f004:**
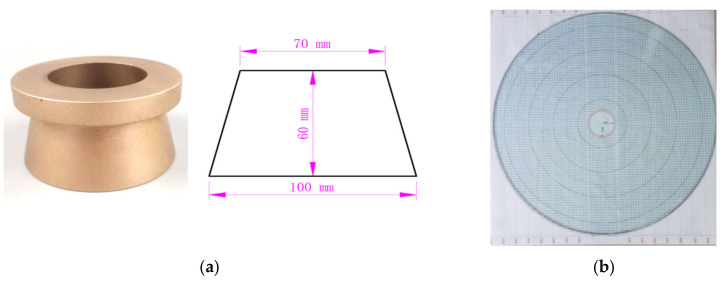
The flow spread test apparatus. (**a**) The frustum cone mold. (**b**) The graduated glass plate.

**Figure 5 materials-13-02939-f005:**
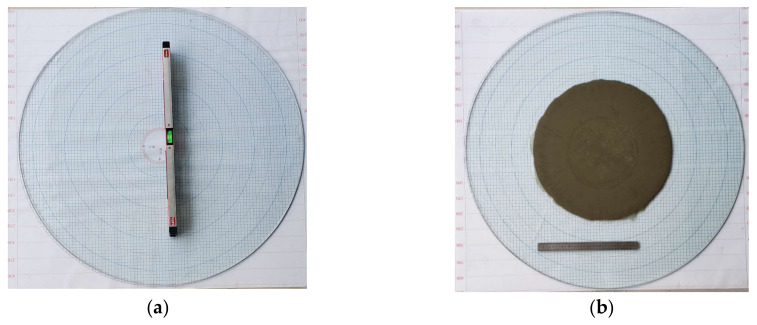
Flow spread testing. (**a**) Glass plate is placed horizontally. (**b**) Grout spreads freely over the glass plate and we can measure the average diameter of the spread.

**Figure 6 materials-13-02939-f006:**
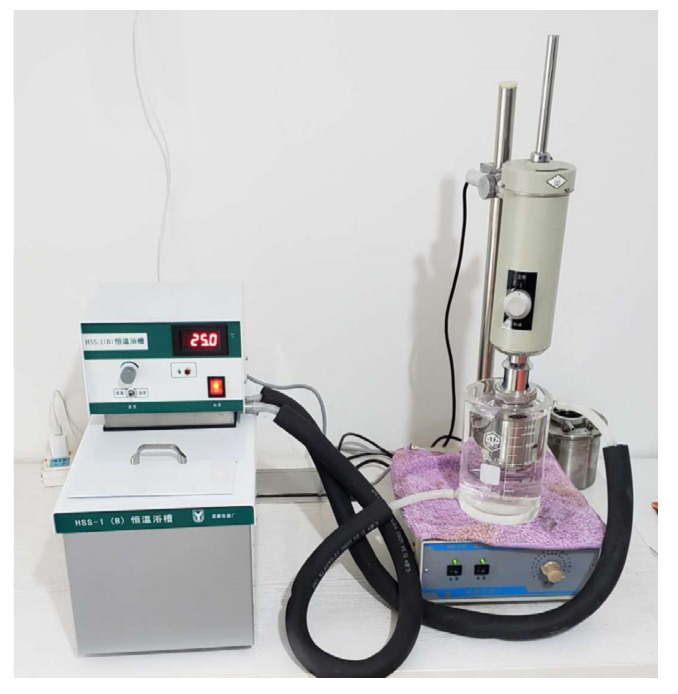
The rotary viscometer.

**Figure 7 materials-13-02939-f007:**
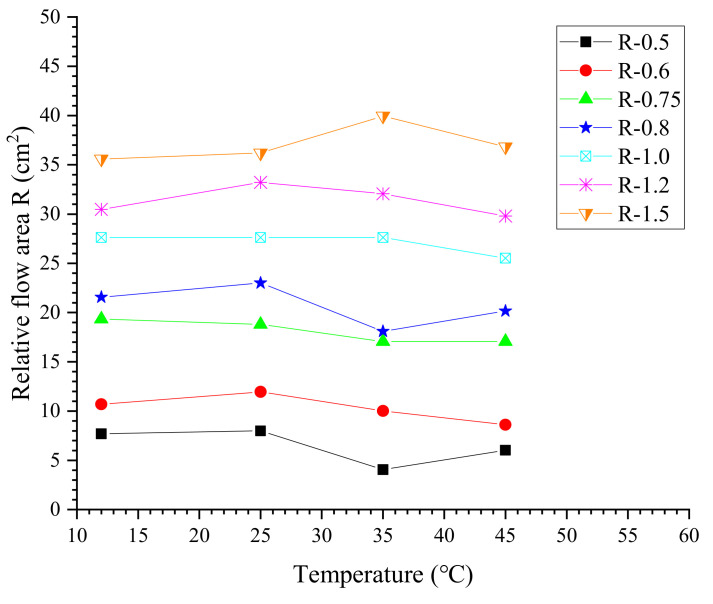
Relative flow area.

**Figure 8 materials-13-02939-f008:**
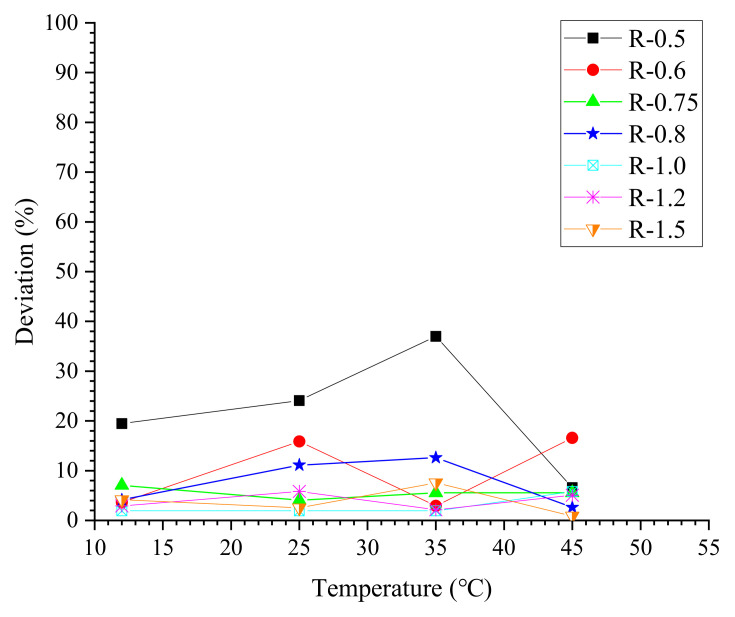
Deviation of measured relative flow area.

**Figure 9 materials-13-02939-f009:**
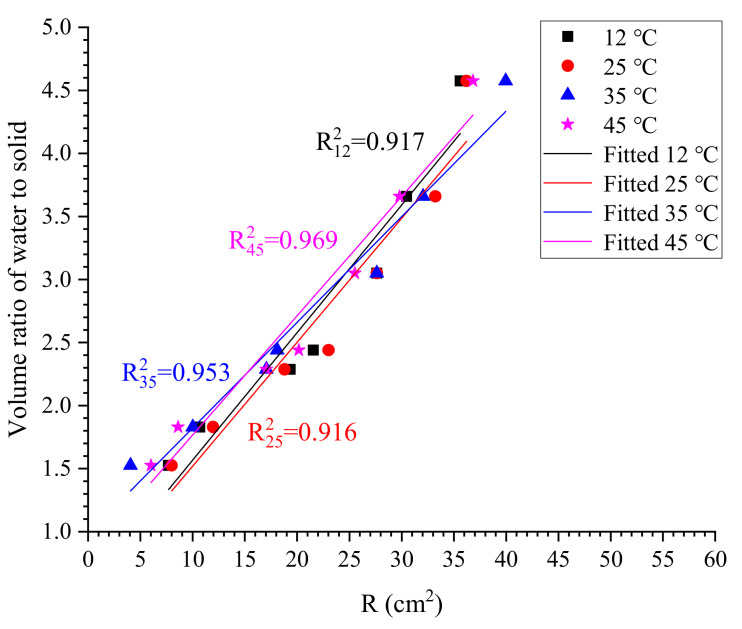
Correlation of R and solid content at 12 °C and 25 °C.

**Figure 10 materials-13-02939-f010:**
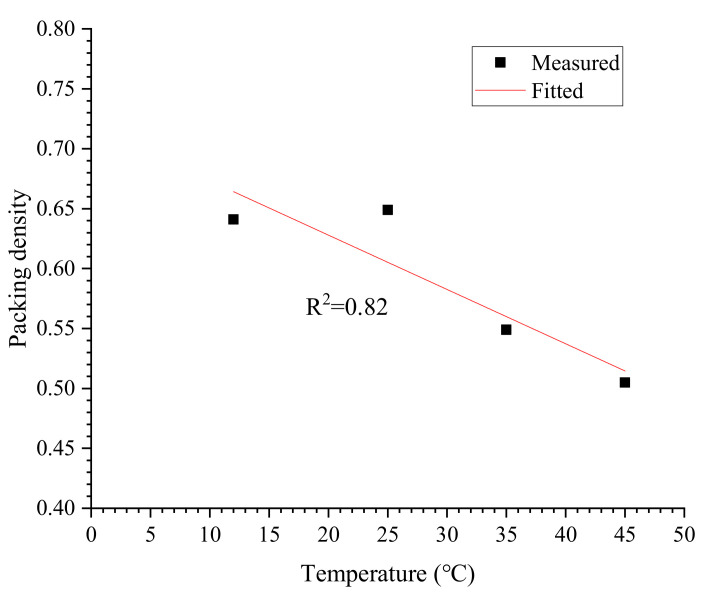
Evolution of packing density versus temperature.

**Figure 11 materials-13-02939-f011:**
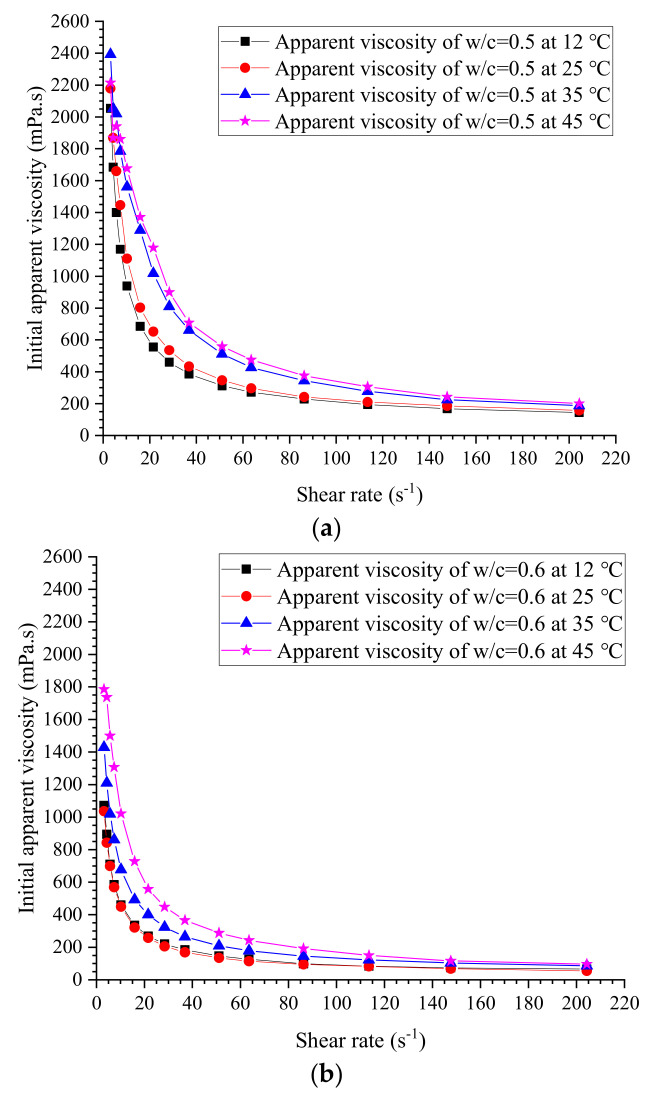
Measured initial apparent viscosity at w/c = 0.5 and 0.6. (**a**) w/c = 0.5. (**b**) w/c = 0.6.

**Figure 12 materials-13-02939-f012:**
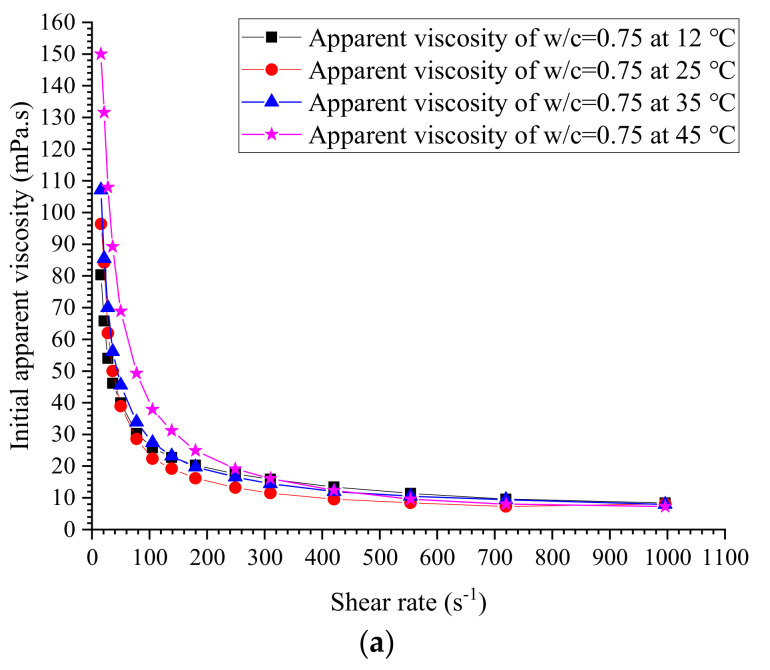
Measured initial apparent viscosity at w/c = 0.75 and 0.8. (**a**) w/c = 0.75. (**b**) w/c = 0.8.

**Figure 13 materials-13-02939-f013:**
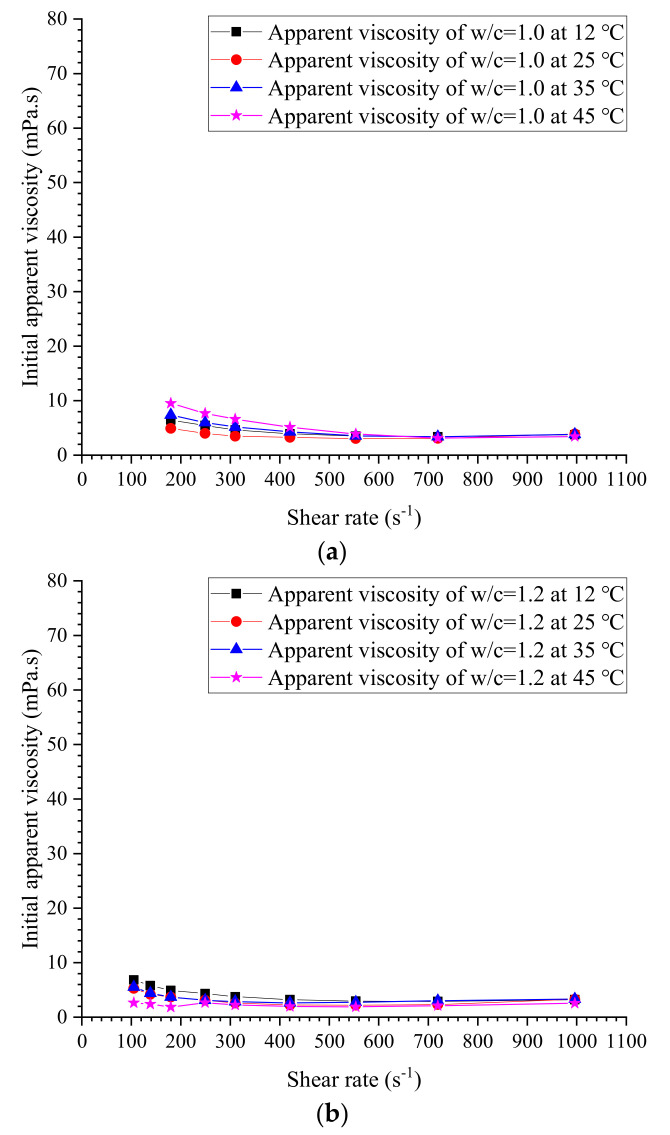
Measured initial apparent viscosity at w/c = 1.0 and 1.2. (**a**) w/c = 1.0. (**b**) w/c = 1.2.

**Figure 14 materials-13-02939-f014:**
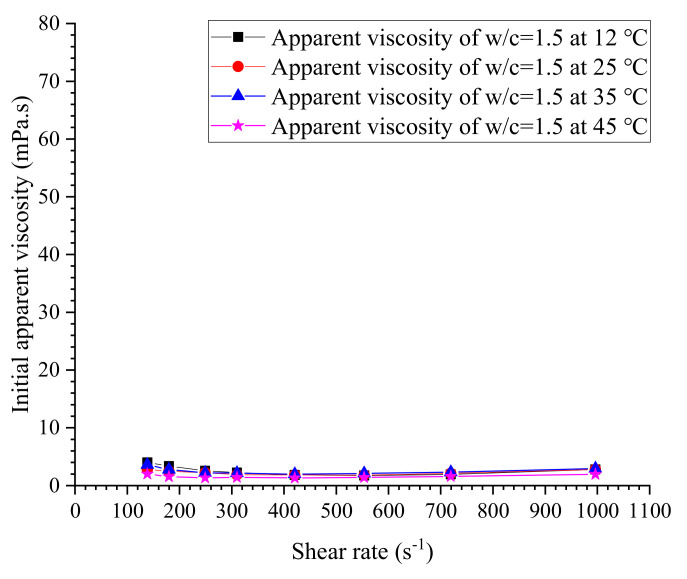
Measured initial apparent viscosity at w/c = 1.5.

**Figure 15 materials-13-02939-f015:**
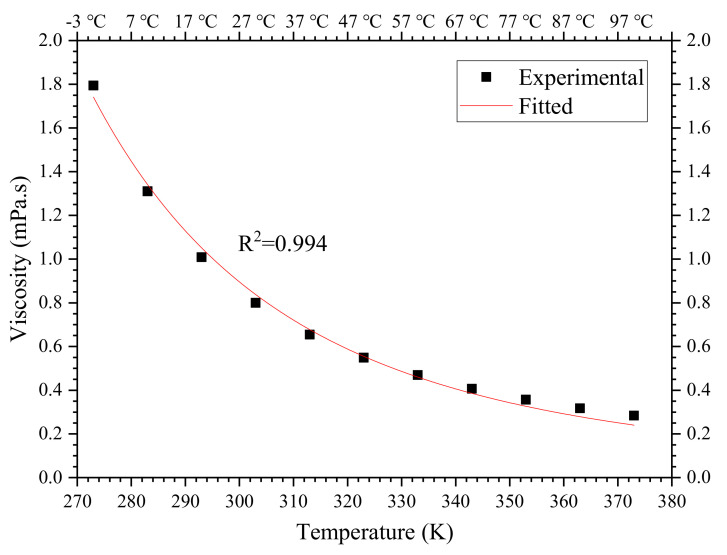
Correlation of water viscosity and temperature.

**Figure 16 materials-13-02939-f016:**
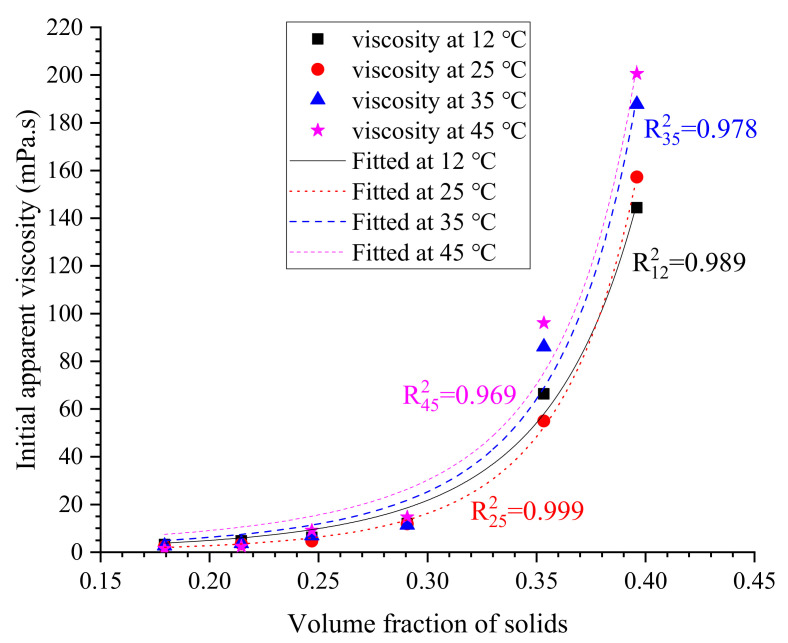
Correlation of grout viscosity and the volume fraction of solids based on Liu’s model.

**Figure 17 materials-13-02939-f017:**
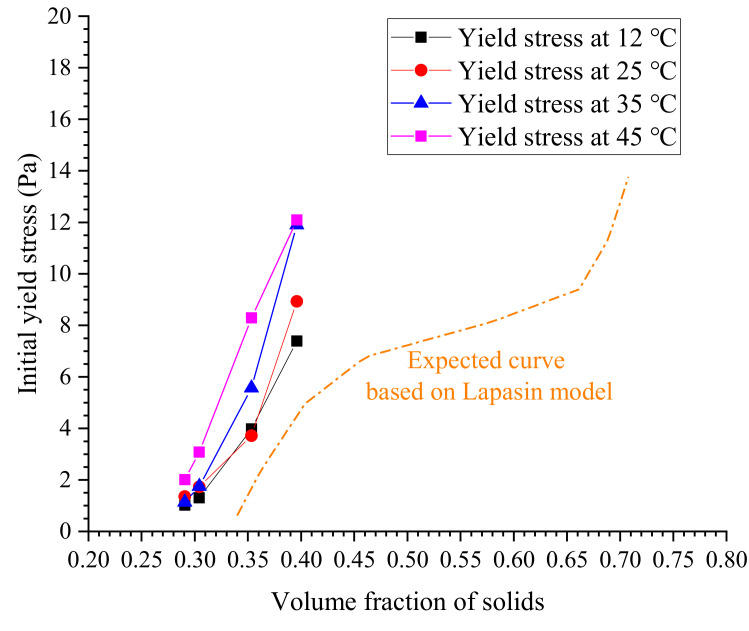
Dependence of initial yield stress on solid content.

**Figure 18 materials-13-02939-f018:**
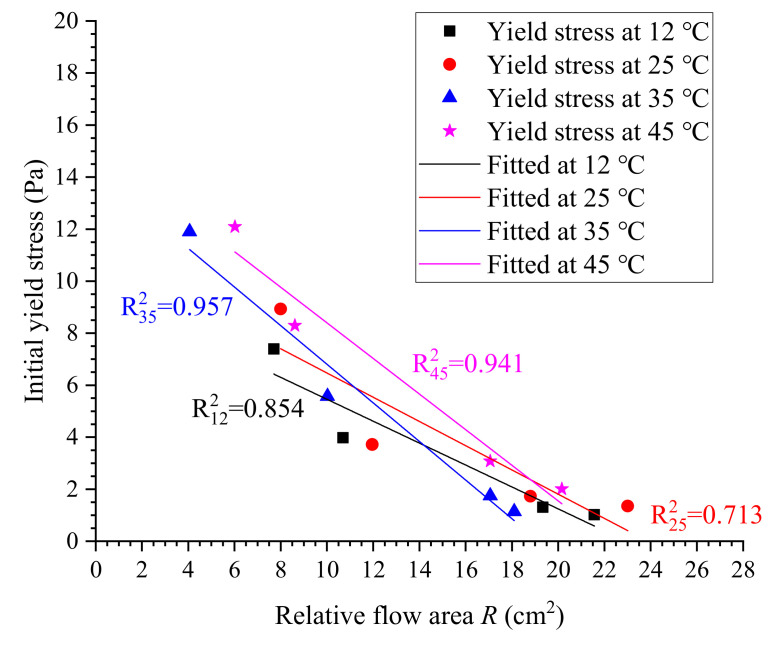
Correlation between initial yield stress and relative flow area.

**Figure 19 materials-13-02939-f019:**
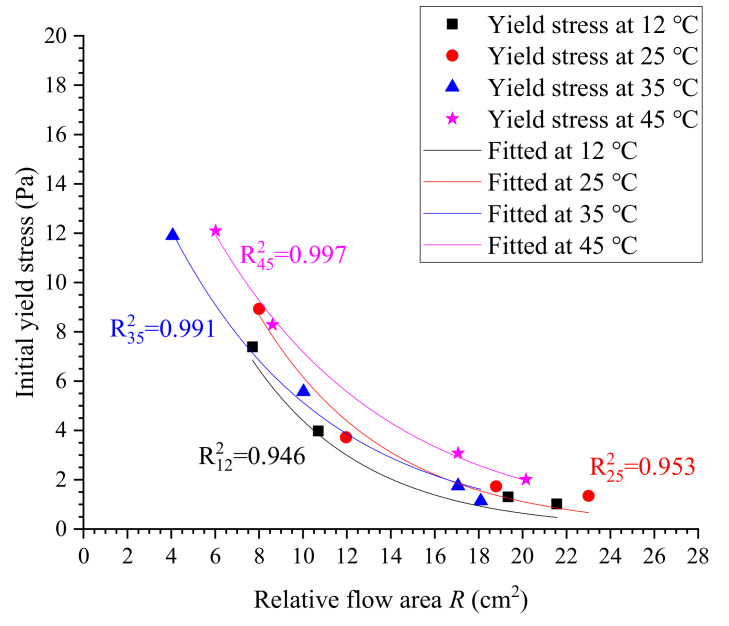
Estimation of yield stress using exponential law.

**Table 1 materials-13-02939-t001:** Properties of the cement utilized.

Chemical Properties		Physical Properties	
Chemical Composition	Amount (wt.%)	Item	Value
CaO	62.47	Specific gravity (g/cm^3^)	3.11
SiO_2_	20.39	Blaine fineness (m^2^/kg)	387
Al_2_O_3_	6.23	Mean grain size D50 (µm)	19.908
Fe_2_O_3_	2.87	Maximum grain size D100 (µm)	81
MgO	1.86	Compressive strength at 3 days (MPa)	25.8
SO_3_	2.95	Compressive strength at 28 days (MPa)	49.5
K_2_O	0.64		
Na_2_O	0.25		
TiO_2_	0.39		
L.O.I.	1.95		

**Table 2 materials-13-02939-t002:** Proportioning of cement grout mixture. TEA: triethanolamine.

No.	Ratio of Water to Cement (w/c)	Dosage of NaCl (wt.% of Cement)	Dosage of TEA (wt.% of Cement)
1	0.50	0.5%	0.05%
2	0.60	0.5%	0.05%
3	0.75	0.5%	0.05%
4	0.80	0.5%	0.05%
5	1.00	0.5%	0.05%
6	1.20	0.5%	0.05%
7	1.50	0.5%	0.05%

**Table 3 materials-13-02939-t003:** Correlation models in the literature [37].

Models	Year	Equation
Mooney’s equation	1951	$\eta_{r}=e^{\frac{\left[\eta \right]\emptyset }{1-K\emptyset }}$
K–D equation	1959	$\eta_{r}={\left[1-\frac{\emptyset }{\emptyset_{m}}\right]}^{-\left[\eta \right]\emptyset_{m}}$
Chong et al.’s model	1971	$\eta_{r}={\left[1+\frac{0.75\frac{\emptyset }{\emptyset_{m}}}{1-\frac{\emptyset }{\emptyset_{m}}}\right]}^{2}$
Dabak and Yucel’s model	1986	$\eta_{r}={\left[1+\frac{\left[\eta \right]\emptyset_{m}\emptyset }{n\left(\emptyset_{m}-\emptyset \right)}\right]}^{n}$
Liu’s model	2000	$\eta_{r}={\left[a\left(\emptyset_{m}-\emptyset \right)\right]}^{-n}$

**Table 4 materials-13-02939-t004:** Calculated values of water demand at various temperatures.

Temperature	$\mathrm{Value}\;\mathrm{of}\;\mathrm{Water}\;\mathrm{Demand}\;\beta_{p}$	R^2^
12 °C	0.56	0.917
25 °C	0.54	0.916
35 °C	0.82	0.953
45 °C	0.98	0.969

**Table 5 materials-13-02939-t005:** Calculated values of packing density of cement at various temperatures.

Temperature	$\mathrm{Value}\;\mathrm{of}\;\mathrm{Packing}\;\mathrm{Density}\;\emptyset_{m}$
12 °C	0.641
25 °C	0.649
35 °C	0.549
45 °C	0.505

**Table 6 materials-13-02939-t006:** Measured initial apparent viscosity at 200 s^−1^.

Ratio (w/c)	Ratio of Water to Solid (V_water_/V_powder_)	Volume Fraction of Solid	Temperature (°C)	Measured Viscosity (mPa·s)
0.5	1.52	0.396	12	144.44
			25	157.22
			35	187.78
			45	200.56
0.6	1.83	0.3534	12	66.39
			25	55.00
			35	86.11
			45	96.11
0.75	2.29	0.3042	12	19.3
			25	15.3
			35	22.6
			45	24.7
0.8	2.44	0.2907	12	12.0
			25	11.8
			35	11.4
			45	14.7
1.0	3.05	0.247	12	4.2
			25	4.65
			35	6.9
			45	9.1
1.2	3.66	0.215	12	4.75
			25	3.5
			35	3.6
			45	2.4
1.5	4.58	0.179	12	3.15
			25	2.4
			35	2.6
			45	1.5

**Table 7 materials-13-02939-t007:** Experimental values of initial yield stress (non-Newtonian fluids).

Ratio (w/c)	Ratio of Water to Solid (V_water_/V_powder_)	Volume Fraction of Solid	Temperature (°C)	Measured Yield Stress (Pa)
0.5	1.52	0.396	12	7.39
			25	8.93
			35	11.9
			45	12.09
0.6	1.83	0.3534	12	3.98
			25	3.72
			35	5.57
			45	8.29
0.75	2.2875	0.3042	12	1.31
			25	1.73
			35	1.75
			45	3.08
0.8	2.44	0.2907	12	1.02
			25	1.35
			35	1.14
			45	2.01

**Table 8 materials-13-02939-t008:** Experimental viscosity of water [44].

Temperature (°C/K)		Experiment Value (mPa·s)	Temperature (°C/K)		Experiment Value (mPa·s)
0	273	1.794	60	333	0.47
10	283	1.31	70	343	0.407
20	293	1.009	80	353	0.357
30	303	0.8	90	363	0.317
40	313	0.654	100	373	0.284
50	323	0.549			

**Table 9 materials-13-02939-t009:** Calculated values of *a* and *n* at different temperatures.

Temperature (°C)	a	n
12	1.791	5.761
25	1.913	7.056
35	1.718	4.146
45	1.342	3.025

**Table 10 materials-13-02939-t010:** Calculated viscosities of cement grout at w/c = 0.75.

Temperature (°C)	Experimental (mPa·s)	Calculated (mPa·s)	Deviation
12	19.3	23.4	+21.5%
25	15.3	17.6	+15.3%
35	22.6	27.1	+20.6%
45	24.7	32.1	+30.8%

**Table 11 materials-13-02939-t011:** Calculated correlation between yield stress and relative flow area.

Temperature (°C)	Equations	Correlation Coefficient
12	$\tau_{0}=9.673-0.421R$	0.854
25	$\tau_{0}=11.127-0.466R$	0.713
35	$\tau_{0}=14.228-0.742R$	0.957
45	$\tau_{0}=15.243-0.685R$	0.941

**Table 12 materials-13-02939-t012:** Fitted exponential models for estimating yield stress.

Temperature (°C)	Equations	Correlation Coefficient
12	$\tau_{0}=30.17\times 0.825^{R}$	0.946
25	$\tau_{0}=34.08\times 0.843^{R}$	0.953
35	$\tau_{0}=21.47\times 0.867^{R}$	0.991
45	$\tau_{0}=25.71\times 0.88^{R}$	0.997

## Data Availability

The data used to support the findings of this study are included within the article.

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
