# Peer review of "Estimation of Viscosity and Yield Stress of Cement Grouts at True Ground Temperatures Based on the Flow Spread Test"

_materials, 2020, doi:10.3390/ma13132939_

Round 1
Reviewer 1 Report
An interesting paper.
There is significant existing published literature on viscosity and yield stress of cement grouts. Section 1 literature review needs to be strengthened.
Can you please check the English
Reviewer 2 Report
This study investigates the viscosity and yield stress of cement grouts, which are important rheological parameters playing a crucial role in rock grouting. The manuscript has adopted a methodology that is appropriate for the investigation. The results are presented with sufficient amount of data worth publishing in this journal and in the relevant field. This paper, if published, will have significant impact. The language and presentation of the paper are sufficiently good, hence, I recommend the paper be published as it is or with a minor spell check.
Reviewer 3 Report
Please see the file attached.

Reviewer 4 Report
The manuscript Estimation of viscosity and yield stress of cement grouts at true ground temperatures based on flow spread test is well written, however, limited conclusions can be taken from the work presented.
Some points to be clarified:
Authors pointed out in the introduction that “ the ground temperature in underground rock mass will be increased from about 30 ℃ 65 at depth of 800 m to 40 ℃ at depth of 1000 m, and when depth is up to 1500 m the ground temperature will be elevated to about 50.” Why temperature investigated in the manuscript is controlled between 12 ℃ and 45 ℃, and not above 50?
Why high w/c ratios were used (above 1). Is this a current practice?
Why Figure 9 and 10 are not combined? That would allow a better comparison.
Figure 11 seems irrelevant. Or y-axis scale should be changed
The conclusions should be reviewed. Conclusion 3 does not seem to be a conclusion of this work, although authors have presented this information test results on the paper, this is well-known information.
Conclusions are very limited and do not bring impactful contributions to the current knowledge/state of art regarding viscosity of grout cements.
Reviewer 5 Report
The manuscript is very interesting, it presents the problem of rheological behaviour of cement grouts in conditions of variable temperature. The research described in the article is not only of high scientific value but also practical one. The research program is clearly and legibly presented. Graphic presentation of both research results and their course is at a high level.
Specific comments:
- The introduction could include a broader review of the literature, which is very extensive in the subject of rheological properties of cement paste.
- Line 101: there is "as same as", should be "as the same as".
- Have you considered the impact of grout segregation on test results? How and did the temperature affect the level of grout segregation?
- In Figure 16, add the x axis with the scale in degrees Celsius.
- Line 402 and figure 19: the trend is rather an exponential function (a ^ x when a < 1). The addition of correlation for the exponential function would enrich the analysis of research results.
